# Association between the Presence of Resistance Genes and Sanitiser Resistance of *Listeria monocytogenes* Isolates Recovered from Different Food-Processing Facilities

**DOI:** 10.3390/microorganisms11122989

**Published:** 2023-12-15

**Authors:** Yue Cheng, Zeinabossadat Ebrahimzadeh Mousavi, Vincenzo Pennone, Daniel Hurley, Francis Butler

**Affiliations:** 1UCD School of Biosystems and Food Engineering, University College Dublin, D04 C1P1 Dublin, Ireland; yue.cheng@ucdconnect.ie (Y.C.); zeinabossadat.ebrahimzadehmousavi@ucd.ie (Z.E.M.); 2Department of Food Science and Engineering, Faculty of Agriculture and Natural Resources, University of Tehran, Karaj 77871-31587, Iran; 3Teagasc Food Research Centre Moorepark, Fermoy, P61 C996 Cork, Ireland; vincenzopennone@hotmail.it; 4UCD School of Agriculture and Food Science, University College Dublin, D04 C1P1 Dublin, Ireland; daniel.hurley@ucd.ie

**Keywords:** *Listeria monocytogenes*, resistance genes, sanitizer resistance

## Abstract

Sanitisers are widely used in cleaning food-processing facilities, but their continued use may cause an increased resistance of pathogenic bacteria. Several genes have been attributed to the increased sanitiser resistance ability of *L. monocytogenes*. This study determined the presence of sanitiser resistance genes in Irish-sourced *L. monocytogenes* isolates and explored the association with phenotypic sanitiser resistance. The presence of three genes associated with sanitiser resistance and a three-gene cassette (*mdrL*, *qacH*, *emrE*, *bcrABC*) were determined in 150 *L. monocytogenes* isolates collected from Irish food-processing facilities. A total of 23 isolates contained *bcrABC*, 42 isolates contained *qacH*, one isolate contained *emrE*, and all isolates contained *mdrL*. Additionally, 47 isolates were selected and grouped according to the number and type of resistance genes, and the minimal inhibitory concentration (MIC) of these isolates for benzalkonium chloride (BAC) was determined experimentally using the broth microdilution method. The BAC resistance of the strain carrying the *bcrABC* gene cassette was significantly higher than that of strains lacking the gene cassette, and the BAC resistance of the strain carrying the *qacH* gene was significantly higher than that of strains lacking the *qacH* gene (*p* < 0.05). Isolates harbouring both the *qacH* and *bcrABC* genes did not show higher BAC resistance. With respect to environmental factors, there was no significant difference in MIC values for isolates recovered from different processing facilities. In summary, this investigation highlights the prevalence of specific sanitiser resistance genes in *L. monocytogenes* isolates from Irish food-processing settings. While certain genes correlated with increased resistance to benzalkonium chloride, the combination of multiple genes did not necessarily amplify this resistance.

## 1. Introduction

*L. monocytogenes* is a bacterial foodborne pathogen that can contaminate food products during or after processing [1]. *L. monocytogenes* is pathogenic in humans, causing listeriosis, and is associated with other human illnesses such as bacteraemia, encephalitis, and sepsis [2]. Listeriosis outbreaks have occurred worldwide. An outbreak in South Africa from 2017 to 2018 was the largest listeriosis outbreak to date, with more than 1000 laboratory-confirmed cases and a 42% fatality rate. The most-infected were infants and pregnant women [3]. Poor hygiene practices and the inadequate implementation of standard sanitation operating procedures in the food-processing industry have led to listeriosis outbreaks [4]. *L. monocytogenes* can also be found in difficult-to-clean tools such as slicers, food transport vehicles, and refrigeration units [5]. The foods most commonly identified as vehicles of *L. monocytogenes* transmission include unpasteurised milk and dairy products, soft cheese varieties, ready-to-eat food, sliced meats, refrigerated smoked seafood, and refrigerated meat spreads [6,7,8,9,10]. In recent years, several novel food vehicles have been implicated in listeriosis outbreaks, such as Enoki mushrooms and packaged salads [11].

The proper cleaning of equipment and processing facilities is of paramount importance to the food industry to control *L. monocytogenes* that may enter the food chain and cause food outbreaks. In a normal sanitation cycle in the food industry, a cleaning agent is applied and rinsed off with water before a disinfectant is applied. After a specific exposure time, the disinfectant is rinsed off with water [12]. Quaternary ammonium compounds (QACs) are cationic membrane-active antibacterial agents [13] that are widely used in the food industry and are known to be effective against *L. monocytogenes* [14]. Benzalkonium chloride (BC) is a commonly used active ingredient in QAC sanitisers. BAC has a broad-spectrum biocidal activity and remains stable for both short- and long-term usage [15].

However, the persistent use of sanitisers may induce the resistance of *L. monocytogenes* towards QACs as a result of selection or adaptation through regular exposure to sublethal concentrations [16]. According to published research, efflux pumps are an important mechanism for *L. monocytogenes* resistance to sanitisers [5,17]. Currently, *L. monocytogenes* have been shown to possess several genetic determinants related to efflux pumps that enhance their tolerance towards QACs [18]. The multidrug efflux pump gene *mdrL* in *L. monocytogenes* was one of the first genes that was confirmed to have a relationship with its resistance to QACs [19]. The *mdrL* gene is located on the chromosome and is present in almost all *L. monocytogenes* serotypes [20,21]. A resistance cassette, known as *bcrABC*, has also been reported to be related to the sanitiser resistance of *L. monocytogenes*. The genes *bcrABC* are composed of a TetR family transcriptional regulator (*bcrA*) and two small multidrug resistance genes (*bcrB* and *bcrC*), all essential for imparting BAC resistance [22]. In addition, *qacH*, a gene encoding a small multidrug resistance protein family transporter located at the chromosomally integrated transposon Tn*6188*, has been reported to provide an increased tolerance of *L. monocytogenes* strains to BAC [23]. Recently, it was reported [24] that the novel efflux pump gene *emrE,* located on the LGI1 genomic island, can also increase the tolerance of *L. monocytogenes* to QACs. It was found that after 1 hour of exposure to BAC (10 μg/mL), the expression of the *emrE* gene increased 49.6-fold, and the growth of *L. monocytogenes* lacking the *emrE* gene in BAC was significantly inhibited.

The objective of this research was to determine the presence of several sanitiser resistance genes in Irish-sourced *L. monocytogenes* isolates and to explore the association between environment, the presence of resistance genes, and the sanitiser resistance of *L. monocytogenes* experimentally. 

## 2. Materials and Methods

### 2.1. L. monocytogenes Isolates and Culture Conditions

A total of 150 isolates of *L. monocytogenes,* collected from Irish food-processing facilities (meat, seafood, vegetables, dairy, and mixed food) and clinical environments, were used in this study. All isolates were provided by Moorepark, Teagasc Food Research Centre, Dublin, Ireland. There were 12 serotypes of *L. monocytogenes* isolates, including 1/2a, 1/2b, 1/2c, 2a, 2b, 3a, 3b, 3c, 4b, 4d, 4e, and 7. The bacterial cultures were kept in storage at a temperature of −80 °C in tryptone soy broth (TSB, Sigma, Wicklow, Ireland) with 20% (*vol*/*vol*) glycerol. When required for use, the cultures were inoculated into Brain Heart Infusion agar (BHI, Sigma, Wicklow, Ireland) plates and grown at 37 °C for 24 h, and afterwards stored at 4 °C until required.

### 2.2. Occurrence of Resistant QAC Genes in Selected L. monocytogenes Isolates

The resistance genes chosen for this study were *mdrL*, *emrE*, *qacH,* and the three-gene cassette *bcrABC* (Table 1). The sequences of these four resistance genes were downloaded from GenBank. The whole genome sequences of the 150 isolates were submitted to GenBank and are available through BioProject numbers PRJNA699172, PRJNA714047, and PRJNA698557.

### 2.3. Minimum Inhibitory Concentration (MIC) Assays

After the genomic analysis, 47 *L. monocytogenes* isolates were selected from the original 150 isolates to give a range of the sequence type, the number of sanitiser resistant genes present, and the source of isolation for MIC evaluation. The primary criterion for selection is to group strains according to the number and types of resistance genes they carry, and the number of strains in different groups should be as equal as possible. Using BLAST gene alignment results, 3 isolates carrying *bcrABC*, *qacH,* and *mdrL*, 14 isolates carrying *bcrABC* and *mdrL*, 13 isolates carrying *qacH* and *mdrL*, 1 isolate carrying *emrE* and *mdrL*, and 16 isolates carrying only *mdrL* were selected. Among isolates carrying the same resistance genes, the selection criteria were designed to include isolates from as diverse a range of process environments as possible.

The MIC assay was performed according to the broth microdilution method provided by the Clinical and Laboratory Standards Institute (CLSI) [28]. Fifteen isolates were randomly selected pre-experiment to establish the likely range of MIC values. The experimental results showed that the MIC of all isolates was lower than 10 mg/L. Therefore, the serial dilution was changed to a 1–10 mg/L fixed concentration difference (intervals of 1 mg/L).

Benzalkonium chloride (BAC) (Sigma, Wicklow, Ireland) was used as a standard QAC. A concentration range of 1–10 mg/L of BAC (intervals of 1 mg/L) was used for MIC evaluation. Initially, a stock BAC solution with a concentration of 100 mg/L was prepared and maintained in a water bath at 54 °C for 20 min to completely dissolve the BAC. The solution was then diluted with distilled H2O to the desired concentration. The *L. monocytogenes* isolates were streaked out onto the BHI plates and incubated overnight at 37 °C. Then, 2–3 colonies from the overnight cultures were transferred to 6 mL of Tryptone Saline Diluent (TSD). The absorbance of the medium was recorded using a spectrophotometer (Thermo Scientific Orion AQ8000-AquaMate, Waltham, MA, USA) at 625 nm. The optical density (OD) of the samples was further adjusted to 0.08–0.13, which corresponded to approximately 1–2 × 10^8^ CFU/mL. Then, 10 mL of double-strength Tryptic soy broth (dsTSB, Sigma, Wicklow, Ireland) was prepared for all the isolates, and 50 µL of the prepared suspension was inoculated into the double-strength TSB. A total of 100 µL of BAC solution at different concentrations (1–10 mg/L) was subsequently transferred to each well of a 96-well plate, and 100 µL of the double-strength TSB containing the culture was transferred to each well. Cultures of each isolate were performed in triplicate. The plate was then incubated at 37 °C for 24 h and the first concentration at which the isolate did not yield visible bacterial growth was considered as the MIC value.

### 2.4. Genetic Analysis

The presence of the *mdrL*, *emrE*, and *qacH* genes and the three-gene cassette *bcrABC* in the genome of the 150 isolates was identified using BLASTN [29]. Multi-locus sequence typing (MLST) was carried out using the mlst tool [30]. The typing standards were traditional PubMLST typing schemes. Parsnp v1.2 [31] was used to align the core genome of the 13 ST121 isolates tested for the MIC in this study. Isolate F2165-17 was randomly chosen as the reference isolate. The core genome phylogeny and multi-alignments were constructed using Parsnp. The *qacH* gene sequences in the genome of the 13 ST121 isolates tested for the MIC was translated to amino acid sequences using MEGA 11.0.10 [32].

### 2.5. Statistical Analysis

Multiple linear regression was carried out between the MIC values and the presence/absence of the *qacH* gene and the *bcrABC* cassette using the R lm function. An interaction term was included in the analysis. The effects of the *emrE* and *mdrL* genes were not included in the analysis as there was only one isolate detected with the *emrE* gene and all isolates contained the *mdrL* gene. A Chi-square test was carried out using the R Chi-square test function on the MIC values depending on the processing facility type.

## 3. Results and Discussion

### 3.1. Occurrence of Genes Conferring Resistance to QACs in the L. monocytogenes Isolates

Appendix A sets out a full list of all the genes detected in each isolate. All the isolates contained the *mdrL* gene, 23 isolates (15%) harboured the *bcrABC* cassette, 40 (27%) isolates harboured the *qacH* gene, and only one isolate harboured the *emrE* gene (Table 2). A total of 34 STs were present in the 150 *L. monocytogenes* isolates. The *bcrABC* gene cassette was identified in a total of seven STs (ST2, ST5, ST9, ST31, ST204, ST836, and ST132). It was strongly associated with several sequence types (present in 11 out of 13 ST5 isolates; two out of two ST132; five out of five ST836). The *qacH* gene was identified in a total of four STs (ST2, ST121, ST122, and ST132). It was strongly associated with ST121 (present in all 37 isolates) and with ST132 (present in all two isolates). The *emrE* gene was only detected in ST9 (present in one out of three ST9 isolates). Two isolates belonging to ST132 and one isolate belonging to ST2 contained the *bcrABC* gene cassette, the *mdrL* gene, and the *qacH* gene.

A study in 2017 of *L. monocytogenes* isolates found that, among the 101 isolates recovered from Norway meat- and salmon-processing facilities, 22% of the isolates contained the *qacH* gene and 8% of the isolates contained the *bcrABC* gene cassette [12]. Compared with that study, the prevalence of *bcrABC* and *qacH* found in this study was higher. An important reason for the increased prevalence of resistance genes may be horizontal gene transfer (HGT). During evolution, bacteria can acquire genetic material through HGT, which consists of conjugation (requiring cell-to-cell contact between cells), transduction (phage facilitates the transfer of genetic information), and transformation (uptake of free DNA from the environment). HGT is the primary mechanism for the spread of antibiotic resistance in bacteria [33]. Among the four resistance genes mentioned in this study, the Tn6188 transposon on which the *qacH* gene is located is related to Tn554 from Staphylococcus aureus and other Tn554-like transposons found in various Firmicutes [23]. The *emrE* gene shows high similarity and amino acid identity with the drug transporter gene in Desulfitobacterium dehalogenans ATCC 51507 [24]. A study on the *bcrABC* gene cassette found that pLM80-like plasmids containing *bcrABC* can spread in different serotypes of *L. monocytogenes* [22]. Thus, HGT may also be a primary way in which *L. monocytogenes* acquires sanitiser resistance genes. In addition, environmental selection resulting from the continued use of sanitisers may also be responsible for the increased proportion of *L. monocytogenes* carrying resistance genes.

### 3.2. MIC Analysis

#### 3.2.1. Effects of *bcrABC* and *qacH* Genes on BAC Resistance of *L. monocytogenes*

The MIC of the 47 isolates tested varied between 1 and 5 mg/L (Table 3). Figure 1 shows the distribution of the MIC values recorded for the isolates. A previous study [12] reported similar MIC values (2–12 mg/L) for *L. monocytogenes* isolates recovered from Norwegian meat and salmon production sites. Multiple regression indicated that the presence of either the *qacH* gene or the *bcrABC* cassette had a significant (*p* < 0.001) positive effect on the MIC values. This is indicated in Figure 2 and Figure 3, which show the variation in MIC values depending on presence/absence of the *qacH* gene or the *bcrABC* cassette. However, there was a negative interaction effect (*p* < 0.01) between the presence of the *qacH* gene and the *bcrABC* cassette, indicating the possible non-synergistic effect on MIC values arising from the presence of both the *qacH* gene and the *bcrABC* cassette in an isolate. This outcome should be viewed with caution due to the small number of isolates containing both the *qacH* gene and the *bcrABC* cassette. The three isolates that contained both the *qacH* gene and the *bcrABC* cassette had MIC values of 4 mg/L, whereas several of the isolates that contained either only the *qacH* gene or the *bcrABC* cassette had higher MIC values. The 3D plot of the MIC values for all the isolates (Figure 4) gives a visual representation of the multiple regression and the interaction. A Norwegian study [12] of 101 *L. monocytogenes* isolates recovered from meat- and salmon-processing plants also showed increased tolerance to BAC in isolates containing either the *qacH* gene or the *bcrABC* cassette (no isolate was found that contained both the *qacH* gene and the *bcrABC* cassette). Another work [14] assessed the activity of the *mdrL* gene-encoded efflux pump in *L. monocytogenes*, and the authors found that *mdrL* efflux pump activity was reduced in *bcrABC*-positive isolates. Hence, they concluded that the higher MIC values recorded for these isolates was caused by the activity of the *bcrABC* cassette.

#### 3.2.2. Differences in Sensitivity to BAC between Groups of *qacH*-Positive Isolates

All the ST121 isolates from the 150 isolates tested were *qacH*-positive. Among the strains selected for MIC testing, 13 isolates were *ST121,* and so all carried the *qacH* gene. Of these 13 isolates, the MIC of the isolate coded as 1513 was 1 mg/L, which was much lower than that of the other ST121 isolates tested (3–5 mg/L). The phylogenetic analysis (Figure 5) indicated that isolate 1513 was significantly different compared to the other 12 ST121 isolates. In addition, the amino acid sequence for the *qacH* gene was identical for the 13 ST121 isolates, except for isolate 1513. Three single-nucleotide differences were found in the sequence of the *qacH* gene of isolate 1513 (Figure 6). This resulted in the following amino acid differences: Ser/Ala at amino acid positions 60 and 63, and Ile/Leu at amino acid position 94. The same mutation was also found in a previous published study [12]. In addition to the differences at these three positions, that study also found a Cys/Ser difference at amino acid position 42, and the isolates with *QacH* variants harbouring 42 Ser had a higher tolerance towards BC than those with Cys in this position. Since the mutation at position 42 was not found in the isolates in this study, but the resistance of the strains to BAC still varied, the mutations at the other three positions may be also responsible for the reduced sanitizer resistance of ST121 *L. monocytogenes* isolate 1513.

#### 3.2.3. Effect of Process Environment on BAC Resistance of *L. monocytogenes*

The mean MIC values of isolates collected from mixed food-, seafood-, vegetable-, dairy-, and meat-processing facilities were 4 mg/L, 4 mg/L, 3 mg/L, 3 mg/L, and 3 mg/L. The analysis of variance indicated no significant difference in the MIC values of isolates recovered from different processing facilities. However, there were differences in the number of resistance genes carried by *L. monocytogenes* isolates from the different processing environments (Figure 7). The facilities processing either seafood or mixed products had isolates with considerably more occurrence of the *qacH* gene and the *bcrABC* gene cassette. Dairy process facilities were the only locations where the *emrE* gene was recovered.

Previous work has shown that the continuous exposure of bacteria to sanitisers develops sanitiser adaptability and tolerance [34]. The tolerance-enhancing mechanisms identified so far include phenotypic adaptation, gene mutation, and horizontal gene transfer [17,35]. In a separate study [36], the MIC to benzalkonium chloride of 25 *L. monocytogenes* isolates increased five-fold following regular exposure. The growth rate of sanitiser-resistant bacteria can be very rapid and will greatly reduce the effect of sanitisers on bacterial growth [37].

In the food production environment, the recommended concentration of QAC sanitisers is between 200 and 1000 mg/L [38], while the highest MIC value detected in this study was only 5 mg/L, which is considerably lower than the recommended working concentration. There were no reported isolates that showed a tolerance higher than 40 mg/L in previous studies on the BAC resistance of *L. monocytogenes* [12,39]. The question has hence been previously raised as to whether this variation in tolerance level has any practical relevance in the food industry [13]. However, the MIC detection method usually used in studies was cell suspension, in which bacteria are suspended in media and do not form biofilms. The mechanisms of increased tolerance via the formation of biofilms include diffusion limitation, the neutralisation of biocides, and the presence of tolerant dormant cells [40]. It has been reported that *L. monocytogenes* in biofilms were more tolerant to QACs than in suspension [41]. Therefore, the current experimental results may not fully represent the resistance of *L. monocytogenes* to sanitisers in food production environments.

## 4. Conclusions

This study determined the sanitiser resistance of *L. monocytogenes* isolates recovered from different Irish food-processing environments. All the isolates contained the *mdrL* gene, 15% harboured the *bcrABC* cassette, 27% harboured the *qacH* gene, and only one isolate harboured the *emrE* gene. The MIC of the isolates tested varied between 1 and 5 mg/L. Multiple regression indicated that the presence of either the *qacH* gene or the *bcrABC* cassette had a significant positive effect on the MIC values, though overall, the absolute magnitude of the difference was small. There was no significant difference in the MIC values of isolates recovered from different processing facility types. However, there were differences in the number of resistance genes carried by *L. monocytogenes* isolates recovered from the different processing environments.

## Figures and Tables

**Figure 1 microorganisms-11-02989-f001:**
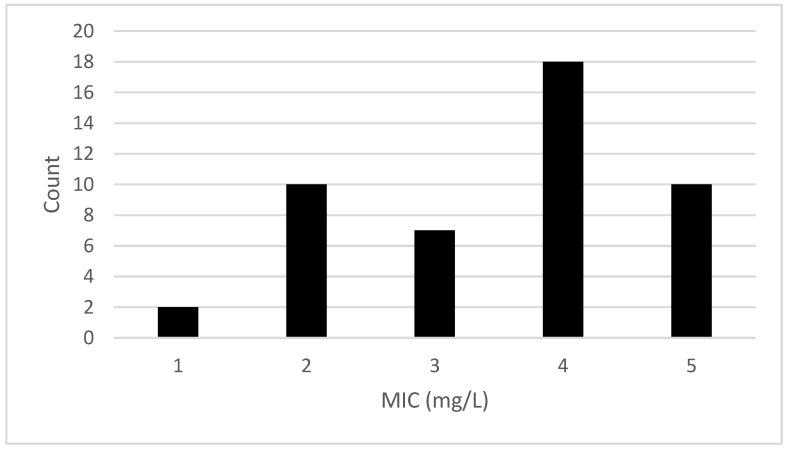
The distribution of MIC values for the 47 isolates analysed in this study.

**Figure 2 microorganisms-11-02989-f002:**
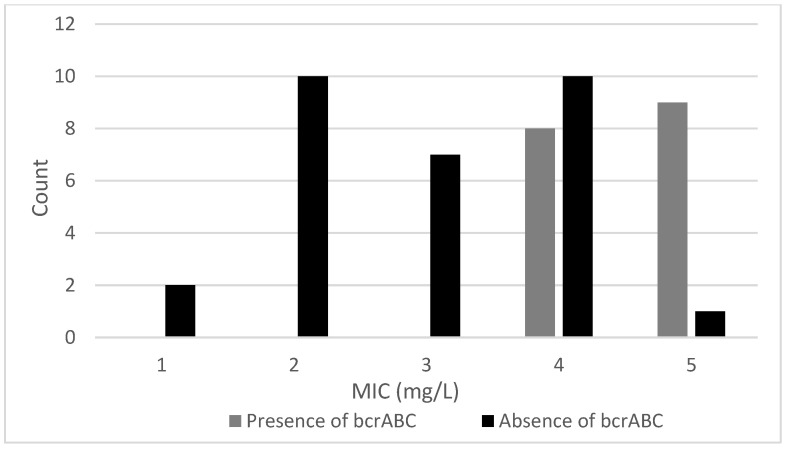
The distribution in MIC values for isolates with or without the *bcrABC* gene cassette.

**Figure 3 microorganisms-11-02989-f003:**
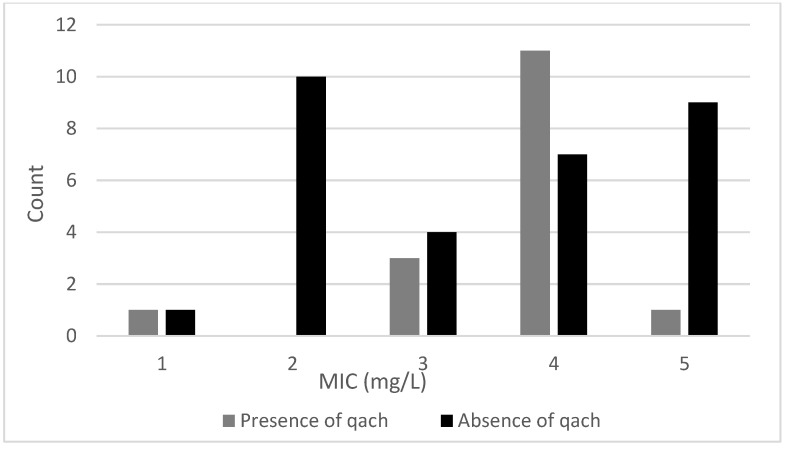
The distribution of MIC values for isolates with or without the *qacH* gene.

**Figure 4 microorganisms-11-02989-f004:**
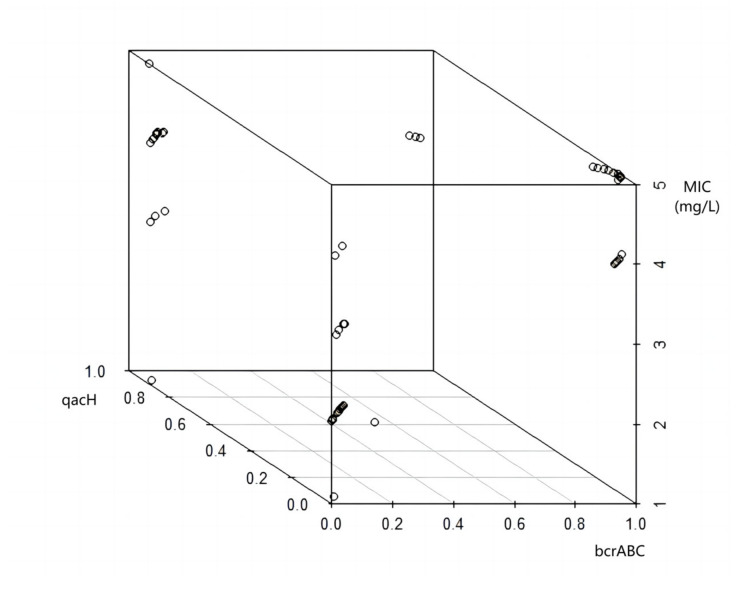
MIC distribution of isolates harbouring either the *bcrABC* gene cassette or the *qacH* gene (presence/absence indicated as 1/0. A small random element has been introduced to distinguish individual test results).

**Figure 5 microorganisms-11-02989-f005:**
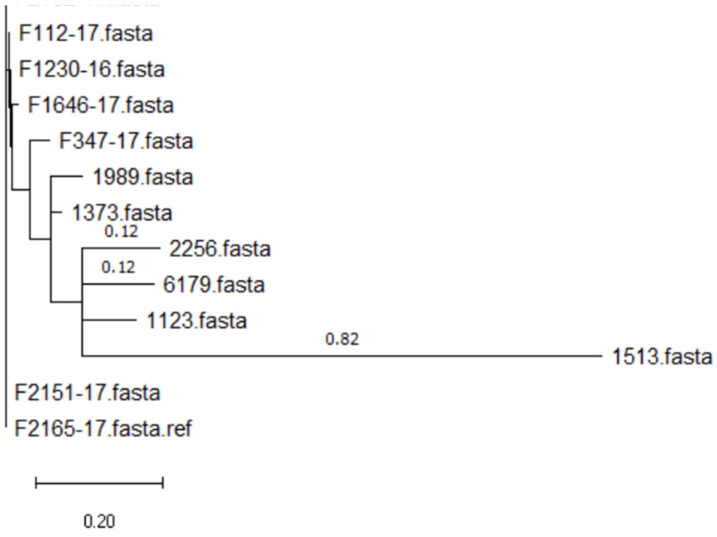
Phylogenetic tree of 13 ST121 *L. monocytogenes* isolates tested for MIC.

**Figure 6 microorganisms-11-02989-f006:**
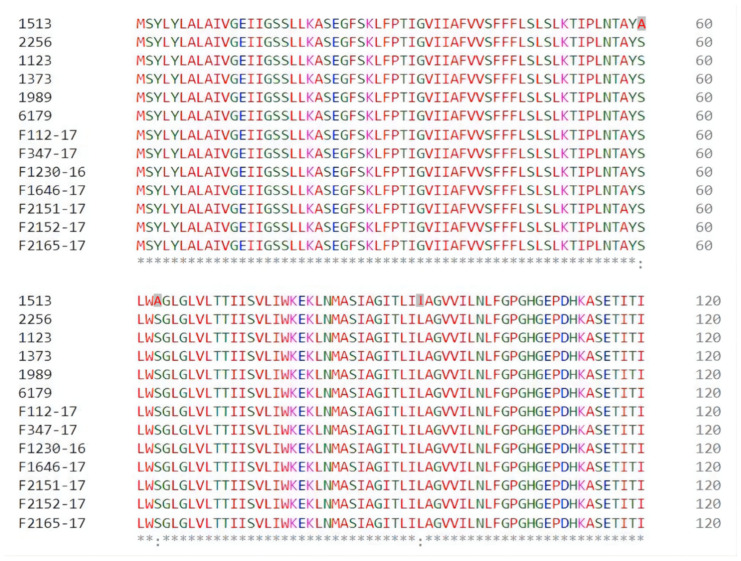
Amino acid sequences encoded in the *qacH* gene of 13 ST121 *L. monocytogenes* isolates (amino acid positions 1–120 are shown). The symbol “*” on the bottom line means there is no difference in the amino acid at this position. The symbol“:” on the bottom line means there is a difference in the amino acid at this position.

**Figure 7 microorganisms-11-02989-f007:**
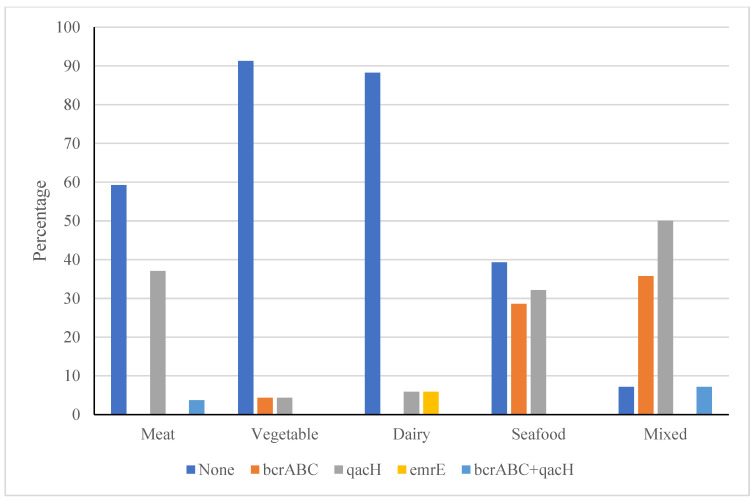
Variation in the percentage of isolates harbouring different resistance genes depending on the process facility.

**Table 1 microorganisms-11-02989-t001:** Overview of QAC resistance genes.

Resistance Gene	GenBank Number	Size	Isolation Source	Reference
*bcrABC*	JX023284.1	1406 bp	Food-processing plant	[22]
*qacH*	MK944277.1	378 bp	Food-processing environment	[25]
*emrE*	CP001602.2	324 bp	RTE food	[26]
*mdrL*	AB671769.1	1194 bp	Yoshihiro Asano Ehime University, Ehime, Japan	[27]

**Table 2 microorganisms-11-02989-t002:** Occurrence of different QAC-resistant genes in the 150 *L. monocytogenes* isolates.

Resistance Gene	*mdrL*	*bcrABC*	*qacH*	*emrE*
Number of isolates	150	23	40	1

**Table 3 microorganisms-11-02989-t003:** MLST profiles, presence of resistance genes, source of isolation, and MIC (BAC) of 47 *L. monocytogenes* isolates.

Isolate No.	CC	ST	Resistance Genes				Source	MIC (mg/L)
			*mdrL*	*bcrABC*	*qacH*	*emrE*		
1564	2	2	+	+	+	−	Meat	4
1387	2	2	+	−	−	−	Seafood	3
2183	2	2	+	−	−	−	Vegetables	2
1413	2	2	+	−	−	−	Vegetables	3
1374	3	3	+	−	−	−	Meat	3
1382	4	4	+	−	−	−	Dairy	2
F1524_16	5	5	+	+	−	−	Mixed food	4
1386	5	5	+	+	−	−	Seafood	5
F347_16	5	5	+	+	−	−	Mixed food	5
F742_17	5	5	+	+	−	−	Mixed food	5
F991_16	5	5	+	+	−	−	Mixed food	5
F1857_15	5	5	+	+	−	−	Mixed food	5
F2166_17	5	5	+	+	−	−	Mixed food	5
F2299_15	5	5	+	+	−	−	Mixed food	5
1392	6	6	+	−	−	−	Seafood	2
1445	7	7	+	−	−	−	Meat	2
1880	8	8	+	−	−	−	Vegetables	2
958	9	9	+	+	−	−	Vegetables	4
1021	9	9	+	−	−	+	Dairy	4
1370	9	9	+	−	−	−	Meat	3
1379	14	14	+	−	−	−	Dairy	2
1389	31	31	+	+	−	−	Seafood	4
1095	37	37	+	−	−	−	Mixed food	2
1006	37	37	+	−	−	−	Meat	1
2256	121	121	+	−	+	−	Vegetables	5
1123	121	121	+	−	+	−	Seafood	4
1373	121	121	+	−	+	−	Meat	3
1513	121	121	+	−	+	−	Seafood	1
1989	121	121	+	−	+	−	Seafood	4
6179	121	121	+	−	+	−	Dairy	4
F112_17	121	121	+	−	+	−	Mixed food	3
F347_17	121	121	+	−	+	−	Mixed food	4
F1230_16	121	121	+	−	+	−	Meat	4
F1646_17	121	121	+	−	+	−	Mixed food	4
F2151_17	121	121	+	−	+	−	Meat	4
F2152_17	121	121	+	−	+	−	Meat	4
F2165_17	121	121	+	−	+	−	Mixed food	3
F113_17	121	132	+	+	+	−	Mixed food	4
F1099_17	121	132	+	+	+	−	Mixed food	4
1385	155	155	+	−	−	−	Seafood	2
1679	204	204	+	+	−	−	Seafood	4
1309	204	204	+	−	−	−	Seafood	4
2226	220	220	+	−	−	−	Vegetables	2
1268	224	224	+	−	−	−	Dairy	2
1306	NOVEL	836	+	+	−	−	Seafood	4
1427	NOVEL	836	+	+	−	−	Seafood	5
1428	NOVEL	836	+	+	−	−	Seafood	5

+ This gene is present in the genetic sequence of this strain. − This gene is not present in the genetic sequence of this strain.

## Data Availability

Additional data presented in this study are available in Appendix A.

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
