# Peer review of "Association between the Presence of Resistance Genes and Sanitiser Resistance of Listeria monocytogenes Isolates Recovered from Different Food-Processing Facilities"

_microorganisms, 2023, doi:10.3390/microorganisms11122989_

Round 1
Reviewer 1 Report
Comments and Suggestions for Authors
The objective of the presented research was to determine the presence of various disinfectant resistance genes in isolates of L. monocytogenes of Irish origin and experimentally explore the association between the environment, the presence of resistance genes, and the resistance of L. monocytogenes to disinfectants.
The following aspects that need improvement and clarification in the manuscript are detailed below:
L19-25. Include the method used for the phenotypic evaluation of BAC within the abstract.
L25-29: The wording is confusing. Refine the wording of L25-27.
L24-29: The conclusion should be regarding its objective, whether it was achieved or not.
L60. According to studies from your group? It is not clear.
L60-61: Add another paragraph regarding the importance of biofilms.
L72-74: Refine the wording, it is redundant.
L74: How does the tolerance to QACs increase? Explain.
L81-82: Delete this information, it is redundant.
L85-92: Considering that these are already isolated strains, more information about them should be provided through a table. Break down sequence types (ST) and isolation locations.
L95 Point 2.2. Why download genes from GenBank for analysis when, in line 142, the authors state that they worked with draft genomes and scanned the genes in PubMLST? In PubMLST Listeria, it is possible to identify 12 loci directly in the genome: Tn6188_tnpA, Tn6188_tnpB, Tn6188_tnpC, Tn6188_qac, Tn6188_tetR, bcrA, bcrB, bcrC, cadA, cadC, LGI-1_LM5578_1862, qacA. Using CARD: qacJ
L105: How were the 47 strains selected? This information is not described in depth: Which sequence types were selected, how many genes were included, and the source of isolation?
L140 and L142-143: What is the reason for duplicating the information? Why align only the 13 ST121 genomes? What happened to the other 44 genomes? Were the genomes submitted to PubMLST Listeria?
L160: A table is required summarizing the characteristics of the 47 studied strains, including ST, Serotype, CC, and cgMLST. The strains not studied can be referenced in a supplementary table.
Table 2 should be rewritten, in line with the requirements in L105. It should incorporate information per strain for the studied genes. What sense does it make to have a table with genes from the unstudied strains?
Table 3 should be placed before point 3 and summarize what is requested.
L160-170: These are only results, there is no discussion.
L180-184: Present these results in a table or figure.
L213-250: Redo figures 1 to 3 to be better understood. For example, Figure 2 discusses the distribution of MIC values from 1 to 5, but in the methodology, it says from 1 to 10 with intervals of 1. Therefore, the figure is not complete. Where is the statistical analysis?
There might be a better way to present Figure 4, as the information needs to be interpreted.
L258-273: These are only results, there is no discussion.
L267-273: What does a mutation at position 42 mean? Why do the authors suggest that this could be responsible for the reduction of the sanitizing effect in the isolate? Present evidence and discuss.
What is the objective of figures 5 and 6? In line 148, the authors mention that the sequences were translated into amino acid sequences; however, they do not mention what type of algorithm was used (NJT, Maximum Likelihood, etc.).
L3.2.3: How will this be done if there is no explicit method for it? How will you statistically adjust the samples if the n's are different, for example, there are only 6 vegetables vs. 12 Seafood vs. 5 dairy vs. 15 mixed food?
L258: Why mention 150 isolates tested when in line 105, it talks about only 47 isolates? This is very confusing.
It is suggested that the manuscript be thoroughly reviewed. It is also recommended to separate Results from Discussion. If you choose to maintain the same format, improve the subtitles for better clarity.
Author Response
Please see the attachements

Reviewer 2 Report
Comments and Suggestions for Authors
The objective of this research is to identify diverse sanitizer resistance genes in Listeria monocytogenes, isolated from a variety of food sectors, and to examine their association with resistance to quaternary ammonium compounds (QAC) disinfectants, specifically benzalkonium chloride. It is important that the title emphasizes the use of QAC or benzalkonium chloride as the disinfectant.
The article is well-structured and written. However, Table 2 might be redundant as the text already provides comprehensive information. Although Figures 2, 3, and 4 are illustrative, they do not offer additional insights beyond what is presented in Table 3.
The one-way analysis of variance (ANOVA) conducted between the R AOV and the MIC may not be suitable for this type of analysis. Given that they are qualitative variables based on frequency, a Chi-square test might be a more appropriate method, even if the final results could be similar.
Round 2
Reviewer 1 Report
Comments and Suggestions for Authors
L152-155: integrate into a single sentence
L186-L204: gene names should be italicized, for example L189-L200
L204: Listeria monocytogenes should be italicized
Author Response
L152-155: integrate into a single sentence
this sentence has now been adjusted
L186-L204: gene names should be italicized, for example L189-L200
This has now been done
L204: Listeria monocytogenes should be italicized
This has now been done
AN updated version of the manuscript has been uploaded
